# Are There Effective Interventions to Increase Physical Activity in Children and Young People? An Umbrella Review

**DOI:** 10.3390/ijerph17103528

**Published:** 2020-05-18

**Authors:** Alice Mannocci, Valeria D’Egidio, Insa Backhaus, Antonio Federici, Alessandra Sinopoli, Andrea Ramirez Varela, Paolo Villari, Giuseppe La Torre

**Affiliations:** 1Department of Public Health and Infectious Disease, Sapienza University of Rome, 00182 Rome, Italy; alice.mannocci@uniroma1.it (A.M.); insa.backhaus@uniroma1.it (I.B.); paolo.villari@uniroma1.it (P.V.); giuseppe.latorre@uniroma1.it (G.L.T.); 2Ministry of Health, 00144 Rome, Italy; an.federici@sanita.it; 3Department of Prevention, Local Health Unit Roma 1, 00161 Rome, Italy; alessandra.sinopoli@aslroma1.it; 4College of Medicine, Universidad de los Andes, 11001000 Bogota, Colombia; aravamd@gmail.com

**Keywords:** systematic review of review, health prevention, chronic diseases, physical activity programs

## Abstract

*Background*: Obesity and physical inactivity among children and young people are public health concerns. While numerous interventions to promote physical activity are available, little is known about the most effective ones. This study aimed to summarize the existing evidence on interventions that aim to increase physical activity. *Methods*: A systematic review of reviews was conducted. Systematic reviews and meta-analyses published from January 2010 until November 2017 were identified through PubMed, Scopus and the Cochrane Library. Two reviewers independently assessed titles and abstracts, performed data extraction and quality assessment. Outcomes as level of physical activity and body mass index were collected in order to assess the efficacy of interventions. *Results*: A total 30 studies examining physical activity interventions met the inclusion criteria, 15 systematic reviews and 15 meta-analyses. Most studies (*N* = 20) were implemented in the school setting, three were developed in preschool and childcare settings, two in the family context, five in the community setting and one miscellaneous context. Results showed that eight meta-analyses obtained a small increase in physical activity level, out of which five were conducted in the school, two in the family and one in the community setting. Most promising programs had the following characteristics: included physical activity in the school curriculum, were long-term interventions, involved teachers and had the support of families. *Conclusion*: The majority of interventions to promote physical activity in children and young people were implemented in the school setting and were multicomponent. Further research is needed to investigate nonschool programs.

## 1. Introduction

According to World Health Organization (WHO) childhood obesity is one of the most important public health challenges of the 21st century. In 2016, 41 million children under the age of five were obese with the majority of them living in developing countries [1]. 

Overweight and obesity have been associated with a wide range of health consequences that often do not become visible until adulthood. As such, overweight and obese children are more likely to develop noncommunicable diseases such as diabetes, cardiovascular diseases and metabolic syndrome during adulthood [2,3]. Key components that can prevent obesity include adequate nutrition and physical activity. 

The WHO recommends that young people aged 5–17 years old should practice at least sixty minutes of moderate to vigorous intensity physical activity (MVPA) daily, of which most should be an aerobic activity [4]. However, currently only around 23% of European boys and 14% of European girls aged 13–15 years reported to meet the WHO recommendations. In Italy, only about 11% of students aged 11–15 years met recommended levels of physical activity [5]. While, a lack of physical activity has been linked to other detrimental behavioral outcomes such as episodes of aggression, substance abuse and other health-risk behaviors [6], regular physical activity has been linked to better physical and mental health outcomes [7]. 

Since noncommunicable diseases are largely preventable, actions against obesity and the promotion of physical activity should be a public health priority. This topic has raised great interest in the scientific community. The Lancet, for example, published two series on physical activity in order to promote global surveillance, epidemiological research, interventions and policy strategies. Although many primary studies as well as systematic reviews on physical activity interventions have been published, complete overviews are rare. To our knowledge, the only currently available umbrella review of physical activity interventions among children and young people is focused exclusively on low- and middle-income countries [8]. Therefore, the aim of this umbrella review is to provide a comprehensive overview of published systematic reviews and meta-analyses on the effectiveness of interventions promoting physical activity among children, adolescents and young people.

## 2. Material and Methods

A detailed protocol for the review has been registered with the International Prospective Register of Systematic Reviews (PROSPERO CRD42018088798). The preferred reporting items for systematic reviews and meta-analyses (PRISMA) statement and the guidelines developed by Aromataris and colleagues were followed to perform this umbrella review [9].

### 2.1. Inclusion Criteria

#### 2.1.1. Study Design, Participants 

Systematic reviews and meta-analyses including randomized-controlled trials (RCTs), cluster-randomized controlled trials (c-RCT), controlled trials (CT), quasi-experimental studies and observational studies were considered in this study. Studies not following a systematic review approach, narrative reviews and primary studies were excluded. Reviews were considered if participants were preschool children (3−6 years), children (6−10 years), preadolescents and adolescents (11−19 years) and university students (19−25 years). Studies including interventions focusing on organizations to which young people were attached, as with indirect involvement, were also included. 

#### 2.1.2. Type of Interventions and Outcome Measures

All interventions aimed at increasing the physical activity level among the young were included. The interventions were classified according to settings of implementation as reported by the physical activity guidelines of the Department of Health and Human Services. The Center for Disease Control and Prevention (CDC) guide was not used as it concerned interventions among both young and adults. The interventions were categorized as follows:Preschool and childcare center setting;School setting;Community setting;Family and home setting.

Reviews with the following outcomes were included in the umbrella review:Physical activity level (moderate, vigorous, moderate to vigorous physical activity (MVPA));Body mass index (BMI).

#### 2.1.3. Language

Reviews published in English, Italian and Spanish language were included in this review.

### 2.2. Exclusion Criteria

Reviews not meeting the predefined criteria mentioned above and reviews about physical activity but not including an intervention or with interventions other than physical activity and movement were excluded.

#### Search Methods for Identification of Studies

Relevant systematic reviews and meta-analyses according to inclusion criteria were identified through systematic searches of the following electronic databases: PubMed, Scopus and the Cochrane Library. The full search strings can be found in Table 1. Studies published from January 2010 until November 2017 were included. Reference lists of identified studies were checked. Primary authors were contacted if clarification or additional information was needed.

### 2.3. Data Collection and Analysis

The first selection was performed filtering duplicate articles using the citation software Zotero 5.0. Eligible studies were selected through a multistep approach (title reading, abstract and full-text assessment). Two researchers (A.M. and V.D’E.) independently analyzed titles and abstracts. Then, each investigator evaluated full texts according to the inclusion criteria. Disagreements between the reviewers were resolved during consensus session with a third reviewer. Where there was ambiguity in trial reporting or lack of data, primary authors were contacted for clarification. 

### 2.4. Data Extraction and Management 

Data were independently extracted by two reviewers (A.M. and V.D’E.) and the following information was considered for each article: First author and year of publication;Title;Study designs (of studies included in the systematic review or meta-analysis);Number of studies included (of studies included in the systematic review or meta-analysis);Brief description of the intervention;Outcomes: physical activity level and BMI;Participants’ age;Length of follow-up;Results: qualitative synthesis of results, odds ratio (OR), standard mean difference (SMD), number of effect sizes (k), effect size Hedge (g) were collected if available;Setting: school, community, family and home, childcare centers.

### 2.5. Quality Assessment Tools

Since flaws in the design, conduct, analysis, and reporting of studies can cause the effect of an intervention to be under or overestimated, two independent reviewers (A.M. and V.D’E.) extracted data on the quality of evidence as well as on the risk of bias. The updated AMSTAR 2 version for systematic reviews and meta-analyses was used to evaluate the methodological quality and risk of bias of studies included in the systematic review [10]. The overall final rate of each systematic review was judged as high, moderate, low or critically low. In case of disagreements, a consensus session with the third reviewer (I.B. or G.L.T.) was held.

## 3. Results

The electronic search initially resulted in 1195 citations. A total of 1121 studies were excluded after the title and abstract screening and 49 full-text articles were selected and read. A total of 30 papers were included in this umbrella review, of which 15 were systematic reviews and 15 were meta-analyses. Appendix A shows the PRISMA flowchart and the study selection process. Characteristics of included systematic reviews and meta-analyses are shown in Appendix A, quality assessment is reported in Appendix A. 

### 3.1. Preschool Setting

Three studies concerned interventions developed in preschool and childcare settings. According to Finch et al. center-based childcare interventions significantly improved physical activity among children (SMD = 0.44, (95% CI: 0.12 to 0.76) [11]. Moreover, significant effects were described for interventions that included a structured activity (SMD = 0.53; 95% CI: 0.12 to 0.94), that were delivered by experts (SMD = 1.26; 95% CI: 0.20 to 2.32) or theory driven (SMD = 0.76; 95% CI: 0.08 to 1.44). Mehtälä et al., who included 23 studies in his systematic review, found a modest increase in the physical activity level of young children who took part in interventions implemented in childcare settings [12]. The systematic review of Ling and colleagues found that theory-driven, multicomponent interventions and interventions involving both parents and their children were most promising [13]. These three studies were judged to be of very low quality according to AMSTAR 2.

### 3.2. School Setting 

A total of twenty studies included in this review focused on the school setting. The studies are clustered according to age groups.

#### 3.2.1. Children: 6−10 Years

Seven studies investigated interventions implemented in the primary school setting for children between 6 and 10 years old. 

Mei et al. [14] found that children’s BMI, who participated in school-based programs, significantly decreased (SMD = −2.23; 95% CI: −2.92 to −1.55) [14]. Similar findings were also noted by Waters and colleagues (SMD = −0.15; 95% CI: −0.21 to −0.09) [15]. In contrast, in the study of Oosterhoff and colleagues, the participation in school-based lifestyle interventions showed no statistically significant results on BMI, (OR = −0.054; 95% CI: −0.13 to 0.02) [16].

The study of Williams and colleagues showed that single interventions had an insufficient effect on the physical activity level of children, while multicomponent interventions, including elements related to both diet and physical activity, showed effective results [17]. Additionally, García and colleagues studied interventions to promote physical activity and healthy nutrition. More than half of the interventions analyzed obtained positive changes in BMI and almost all showed an increase of physical activity level [18].

The systematic review of Wolfenden et al. studied multicomponent interventions that promoted a healthy lifestyle using a combination of strategies: the diffusion of educational materials, educational meetings, incentives or grants to increase teachers’ knowledge, skills or attitudes [19]. Five of the eight trials reported significant improvements in physical activity. Watson et al. found that classroom-based physical activity did not have significant effects for physical activity levels (SMD = 0.40; 95% CI: −1.15 to 0.95) but had a positive effect on improving on-task and reducing off-task classroom behavior [20].

According to the AMSTAR 2 assessment, the studies of Waters et al. [15] and Wolfenden et al. [19] yielded a high-quality score. The review by Oosterhof et al. [16] was rated as low quality. All other studies concerning intervention for children aged 6 to 10 years were evaluated of critically low quality. 

#### 3.2.2. Preadolescents and Adolescents: 11–19 Years

Out of twenty studies, three studies dealt with interventions implemented in secondary schools. Borde and colleagues who conducted a meta-analysis found that the pooled effects for both total physical activity (SMD = 0.02, 95% CI: −0.13 to 0.18) and MVPA (SDM = 0.24, 95% CI: −0.08 to 0.56) were not significant [21]. Age of the sample was a significant moderator for total physical activity, with a younger age associated with a larger effect [21]. Owen et al. found in their systematic review and meta-analysis that school-based interventions produced a small, but significant positive effect on physical activity levels among adolescent girls (k = 16, g = 0.07, *p* = 0.05) [22]. The study of Waters and colleagues found that school intervention for students 13−18 years old did not have a statistically significant effect on BMI (SMD = −0.09, 95% CI: −0.20 to 0.03) [15].

The studies conducted by Owen et al. [22] and by Borde et al. [21] were considered to be of low to very low quality, respectively; whereas the study by Waters et al. [15] was considered to be of high quality.

#### 3.2.3. College and University Students: 20−25 Years

One study dealt specifically with university and college students [23]. Plotnikoff and colleagues studied university lifestyle interventions and found significant results in the intervention group compared to the control group in increasing levels of moderate physical activity (SMD = 0.18; 95% CI: 0.06 to 0.30), but no significant results for vigorous physical activity [23]. One included study, for example, assessed the influence of a single-semester university general education health and wellness course on physical activity and dietary habits among university students. The authors found that students overall level of physical activity improved by 12% [23]. Another included study evaluated the effectiveness of a theoretically based and Web-delivered intervention using common course technology for increasing physical activity in a college student sample and found that participants reported increased days of moderate and vigorous physical activity [23]. According to the AMSTAR 2 assessment the study is of critically low quality.

#### 3.2.4. Intervention for Both Children and Adolescents

Eight studies considered interventions in both primary and secondary school settings for children and adolescents. The reviews of Lonsdale et al. [24] and Dobbins et al. [25] investigated interventions conducted in primary or secondary schools to improve time spent in MVPA during physical education lessons. Both studies showed higher physical activity levels in the intervention groups compared to the control groups (SMD = 0.62, 95% CI: 0.39 to 0.84) and (OR 2.74; 95% CI: 2.01 to 3.75), respectively. [24,25]. Gorga and colleagues [26] showed a significant improvement in BMI in all eleven projects that promoted a healthy lifestyle among children and adolescents [26]. Camacho-Miñano and colleagues [27] studied the effect of interventions to promote physical activity among girls aged 5−18 years; 10 of the 21 studies reported a favorable effect upon physical activity outcomes [27]. Martin and Murtagh [28] studied classroom-based physical activity interventions that integrated academic content through the use of physically active teaching methods. Out of 15 studies, 6 reported an increase in physical activity levels and three studies showed positive effects on children’s BMI [28]. 

The studies included in this section were classified as very low quality except for Dobbins and colleagues [25], which was of high quality. The review of Russ analyzed multicomponent school physical activity programs and reported a small but significant effect on total daily physical activity (g = 0.11, 95% CI: 0.03 to 0.19) [29]. This study was rated of very low quality.

Atkin and colleagues provided a systematic review of interventions to promote physical immediately after school. The authors showed that out of nine studies three reported an increase in physical activity. However, the authors also stress that difficulties were registered for feasibility and applicability and that limitations in study design, lack of statistical power and problems with implementation have likely hindered the effectiveness of interventions in the after-school setting [30]. This study was considered to be of critically low quality according to the Amstar 2 evaluation. 

### 3.3. Family and Home Setting

Two studies included in the overview were developed in the family and home setting. The meta-analysis by Brown and colleagues summarized 19 studies and found a significant, although small effect, in favor of children involved in family-based interventions (SMD: 0.41; 95% CI: 0.15 to 0.67) [31]. Dellert and Johnson investigated the effect of interventions including parents and children on children’s physical activity and body mass index. The authors found that children’s physical activity improved significantly when using a combined parent–child intervention (SMD 0.29, 95% CI: 0.09 to 0.48), but not BMI: (SMD −0.09, 95% CI: −0.37 to 0.19) [32]. Both studies were considered to be of low quality according to Amstar 2 evaluation.

### 3.4. Community Setting and Miscellaneous Context

One study identified programs implemented in the community setting [33] and one study was considered to be miscellaneous [34]. Metcalf and colleagues analyzed in their review 30 RCTs or CTs about community interventions with a component to increase the physical activity in children and adolescents. The overall effect was small for total physical activity (SMD: 0.12, 95% CI: 0.04 to 0.20) and MVPA (SMD: 0.16, 95% CI: 0.08 to 0.24) [33]. Stratification for age categories did not show differences among groups. This study was evaluated as very low quality. The systematic review of Bleich et al. [34] was considered to be miscellaneous as it studied interventions in different settings. School-based interventions with a multicomponent approach had the greatest effectiveness, while clear conclusions cannot be drawn for other settings (preschool, community and home-based interventions) due to paucity and heterogeneity of studies [34].

### 3.5. e-Health Interventions

Four studies focused on e-health interventions to promote physical activity [35,36,37,38]. Schoeppe and colleagues examined the efficacy of interventions using apps: half of the interventions reported no significant changes while the other half reported significant changes [35]. Lau et al. [36] studied interventions employing communication technologies such as email or text messaging to increase physical activity: out of nine RCTs, seven demonstrated efficacy in at least one psychosocial (e.g., physical activity intention) or behavioral (e.g., self-reported physical activity) physical activity outcome. Positive effects were increased when combing it with an intervention such as a face-to-face approach [36]. The authors highlight, however, that even though most included studies demonstrated satisfactory methodological quality, several quality criteria should be considered in future studies. Among others these include a clear description of allocation concealment and blinding of outcome assessment [36]. Furthermore, Hamel and colleagues [37] studied e-health interventions and found that the majority of interventions resulted in positive changes in physical activity levels. However, the authors also noted that interventions that included post intervention follow-up, ranging from 3−18 months, showed that these changes were not maintained [37]. Also, McIntosh and colleagues [38] assessed in their systematic review the effectiveness of e-health interventions on increasing physical activity levels. The researchers found that e-health interventions that incorporated the social cognitive theory were successful and significantly increased physical activity levels [38]. According to the AMSTAR 2 assessment the studies were of critically low to low quality.

### 3.6. Quality of Included Reviews

The methodological quality of included reviews was generally low. Three studies were judged to be of high quality, 3 of low quality and 24 studies of critically low quality. Most studies failed to satisfy item 7 (i.e., list of excluded studies). Besides the studies conducted by Wolfenden [19] and Waters [15], all other studies failed to report AMSTAR item 10 (i.e., sources of funding). The systematic review conducted by Garcia and colleagues [18] failed to meet nearly all items, besides AMSTAR Item 5 (i.e., study selection in duplicate). Findings of the quality assessment are presented in Appendix A.

## 4. Discussion

The purpose umbrella review was to summarize and comprehensively review systematic reviews and meta-analyses on the effectiveness of interventions promoting physical activity among children, adolescents, young people. A total of 30 studies, which focused on physical activity interventions implemented worldwide were included. The setting were most interventions were implemented was the education setting (i.e., primary and secondary schools and university). Our ability to draw adequate conclusions was limited because either the quality of the systematic review or meta-analysis was poor or because, as noted in several reviews, the quality of primary studies was poor. Nonetheless, a few interventions seem more promising in increasing physical activity than others.

For preschool children interventions that included a structured activity, that were delivered by experts, theory driven and/or involved both children and parents were most effective. For university students, it appeared that overall the tertiary education setting is an appropriate settings for implementing physical activity interventions. Both classroom and web-based interventions were successful in increasing physical activity, especially if an intervention was based on the social cognitive theory. 

The present study found mixed results for school-based interventions. They seemed to be effective if they use a multicomponent approach, are embedded into school curriculum and include teacher, parent and student physical activity education [11,28]. This is in line with previous reviews that showed that programs combining diet and physical activity programs produced greater beneficial effects compared to single strategies [16,17,19]. Others key components of effective programs were long-term interventions [33,39], the participation of teachers and the involvement of parents [16,40]. Parental involvement is often an integral component of school-based interventions with strong evidence indicating that interventions involving schools in combination with family members increase physical activity level in children [15]. Concerning the length of interventions, only nine studies had a follow-up of six months or longer, although outcomes were not stratified according to follow-up periods due to the heterogeneity of the results. Furthermore, the effectiveness appears to vary by age. While interventions focusing on primary schools obtained small but significant effects on physical activity levels and on BMI, interventions applied in secondary schools, among adolescents, showed only a very small or nonsignificant effect on physical activity levels [15,21,22]. The results are in line with previous research that suggested the effects of school-based interventions targeting adolescents are questionable [25,33].This may be due to the complex physiological and psychological changes that take place during adolescence, which can determine difficulties to obtain behavioral changes [22]. Nonetheless, schools remain an important setting for physical activity promotion, given that no other institution has more contact with young people during the first two decades of life. In addition, as stated by McIntosh [38], it is important to note that the ability of young people to achieve and sustain an increase in physical activity may be largely determined by their social environment and their social support. If no one walks to school, it is unlikely for someone to begin walking to school. Furthermore, Camacho et al. highlighted in their review that adolescents are highly conditioned by the behavior of their peers [27].

This umbrella review identified few reviews that studied the effect of interventions in childcare and nonschool settings. Interventions implemented in the preschool setting were very few and obtained a good effect on physical activity levels [11,12,13]. Most promising programs included characteristics such as the structured physical activity component, the use of behavioral change theory in intervention design, and the involvement of both parents and children [11]. Nevertheless, considering the paucity of results, firm and definite conclusions about preschool interventions cannot be drawn. The context of the family showed little positive results, but studies were very few and of low quality. According to previous scientific researches, however, the family and home environment are important to determine the health behavior of children and adolescents [28]. In particular, the physical activity level of parents and parental support are known to have a great impact on young people’s behavior [31]. 

This review also identified a small number of studies, mostly of low quality, that focused on e-health interventions and the impact of information and communication technologies (e.g., apps and mobile phones) to promote physical activity among young people. All authors concluded that e-health interventions can be effective in improving physical activity, especially when combined with other intervention strategies [36]. These interventions should be further investigated given that a) young people are digital natives and b) they offer an accessible, cost-effective and time efficient approach for the promotion of lifestyle improvements. Furthermore, a growing area of research has suggested that e-health technologies allow for more individualized behavior change interventions.

With respect to the methodological quality assessment, more than two-thirds of the included studies were classified as having a very low or low methodological quality. In particular, most studies missed item number two, which concerns the presence and registration of the study protocol and item seven, which refers to the inclusion of the list of excluded studies. Furthermore, most articles reviewed did not report sources of funding of their studies, and only half of the studies explained or partially explained the risk of bias (items number 9, 12 and 15) and considered the consequence of these bias in the discussion. These aspects, however, are fundamental for the quality of a review since they strongly depend on the quality of the primary evidence. 

### Strength and Limitations

This umbrella review has strengths and limitations that should be acknowledged when interpreting the results. The main strength of this umbrella review is that it provides a systematic synthesis of the most recent scientific evidence on strategies to promote physical activity among young people worldwide. Additionally, a rigorous quality assessment was performed using the latest version of a measurement tool to assess systematic reviews, namely the AMSTAR 2. Gathering these findings in one place has generated a comprehensive overview of which interventions are most likely to increase physical activity levels in young people. 

Nevertheless, this overview presents some limitations. One of the limitations most commonly cited by review authors includes the quality of primary studies. Moreover, several authors noted an unclear and sometimes even high risk of bias of the primary studies [11,23,36]. If the raw material is flawed, then the conclusions of a systematic review cannot be fully trusted. Furthermore, searches were restricted to articles in English, Italian and Spanish and other potentially relevant articles may have been missed. This overview deeply investigated the effect of interventions on physical activity levels, but other physical, psychological and social outcomes that could influence physical activity were not analyzed. Last, the considerable heterogeneity in study designs included in the single review, types of strategies, components of single interventions, duration, outcomes and measures limited the synthesis of results. 

## 5. Conclusions

In conclusion, interventions that seem most likely to have a positive effect on young people’s physical activity level included interventions with a multicomponent approach. However, given the low quality of included studies drawing concrete conclusion is ambitious. Therefore, further high-quality research that examines the effectiveness of physical activity interventions is required.

## Figures and Tables

**Table 1 ijerph-17-03528-t001:** Search strings.

Database	Search String
PUBMED	(physical activity OR physical inactivity OR sedentary lifestyle) AND (polic* OR health policy OR health promotion OR health impact) AND (adolescent OR young OR children OR youth OR teenager) AND (Review OR systematic OR Metanalysis)
SCOPUS	( TITLE-ABS-KEY ( physical AND activity ) OR TITLE-ABS-KEY ( physical AND inactivity ) OR TITLE-ABS-KEY ( sedentary AND lifestyle ) AND TITLE-ABS-KEY ( polic* OR intervention ) AND TITLE-ABS-KEY ( health OR promotion OR impact ) AND TITLE-ABS-KEY ( adolescent OR young OR children OR youth OR teenager ) ) AND ( LIMIT-TO ( PUBYEAR , 2017 ) OR LIMIT-TO ( PUBYEAR , 2016 ) OR LIMIT-TO ( PUBYEAR , 2015 ) ) AND ( LIMIT-TO ( DOCTYPE , “re” ) ) OR LIMIT-TO ( PUBYEAR , 2014 ) OR LIMIT-TO ( PUBYEAR , 2013 ) OR LIMIT-TO ( PUBYEAR , 2012 ) OR LIMIT-TO ( PUBYEAR , 2011 ) OR LIMIT-TO ( PUBYEAR , 2010 ) ) AND ( LIMIT-TO ( DOCTYPE , “re” ) )
COCHRANE	(physical activity OR physical inactivity OR sedentary lifestyle) AND (polic* OR health policy OR health promotion OR health impact) AND (adolescent OR young OR children OR youth OR teenager) AND (Review OR systematic OR Meta-Analysis) in Title, Abstract, Keywords, Publication Year from 2010 to 2017 in Cochrane Reviews’

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
