# Peer review of "Are There Effective Interventions to Increase Physical Activity in Children and Young People? An Umbrella Review"

_ijerph, 2020, doi:10.3390/ijerph17103528_

Round 1

Reviewer 1 Report

Congratulations for the good reviews.

Actually, the audiences hope to see more illustrations or tables addressing some details, please try to deliver some tables, not only the text, otherwise too heavy. 

Reviewer 2 Report

Overall the analysis performed was done with appropriate methodologies and was adequately described.  I particularly appreciated the discussion of the poor or low quality of many of the studies, unfortunately poorly planned (and reported) studies are all too common.  I don't have any comments on the methods used, however there are several instances where the writing needs to be reviewed by a native English speaker.  I highlight some of those instances below, but there are others.  The sometimes awkward writing makes understanding the point of some sentences difficult, and the concepts could be much more clearly stated.  In addition, there seem to be things missing from the manuscript (i.e. supplementary figure 1, heading of columns in supplemental table 3).

If a close editing of the writing and the addition of the missing parts is completed, it will be a valuable umbrella review of physical activity interventions in children and the young.

Specific examples that need attention are listed below by page and line number where applicable:

Page 1, Line 40: The following sentence, “Instead key components in the fight against obesity are good nutrition and physical 41 activity” doesn’t make sense to me as a reader. Possibly the word “instead” should be removed?  Or the sentence should be reworded?

Page 2, Line 47: The following sentence, “Whereas, a lack of physical activity has been linked to other behavioural outcomes such as episodes of aggression, substance abuse and other health-risk behaviours [6] while regular physical activity has been linked to numerous physical and mental 49 benefits [7].” Should remove the word “while”. It’s very awkward as written.

Page 4; line 132: The PRISMA flowchart was described as being shown in Supplemental Figure 1, however no figures were included in the Supplemental materials.  I think the PRISMA flowchart is an important figure for the reader to see to assess the number of manuscripts that did not meet criteria for inclusion here.

Page 4; Line 140: “More than half studies of the systematic review of Mehtälä et al. found a modest increase in level of physical activity in young children for intervention in childcare setting [12].” The first part of this sentence seems to be missing some words, or it needs to be reorganized.

Page 4; line 169: “behaviuor” is misspelled.

Page 5; line 178: The following sentence is missing something “The study of Owen et al. showed a significant but small positive effect for school-based interventions to increase level of physical activity in adolescent girls very small positive effect (k = 16, g = 0.07, p = 0.05) [22].”  What does “physical activity in adolescent girls very small positive effect” mean?

Page 6; Line 258: The authors state “This finding may be due to the complex physiological and psychological changes that take place during adolescence, which can determine difficulties to obtain behavioural changes [22].” What do the authors mean by saying the changes during adolescence can “determine” difficulties in affecting behavioral changes?

Page 6; Line 260: “Camacho et al. highlighted that adolescents' is highly conditioned by the behaviour of peers although this mechanism could be used in health promotion programs through peer-based strategies [26].” Are the authors saying the “adolescents are” highly conditioned by behavior?  Or that the time of “adolescence” is highly conditioned by behavior?

Page 7; line 290: The authors state “This overview obtained good but few results about e-health interventions in children and adolescents…” . What is meant by “good”?  When including a subjective assessment like this, there needs to be a qualifier as to how the authors defined good.

Supplemental table 3 seems possibly to be missing the table headers. The table lists the quality assessments (numbered 1 through 16) of each manuscript included in the analysis; however it does not define what the assessments are.  I assume the table header described what quality assessment number 1 is and so on, but without the heading, it is impossible to tell.  As this is important information (i.e. allows the reader to look at the included manuscripts and determine the rating for each type of quality assessment), please make sure it is included.

The footnote to supplement table 3 defines “1”, “2”, of “/”. It does not define “N/A” and “0”, both of which are given in the table.  “N/A”, while self-explanatory, should still be defined in the footnote.  I do not know what “0” indicates?

Reviewer 3 Report

From my point of view the work can be considered ok as it is. The only concern is about the lack of some information in the supplementary material. In fact was not possible to find Supp. Figure 1 and its caption. Moreover, information provided by the Tables should be described in detail. For instance parameters from 1 to 16 should be detailed, otherwise the whole Supp. Table 3 was meaningless.
